# Partial inhibition of class III PI3K VPS-34 ameliorates motor aging and prolongs health span

Zhongliang Hu[1]☯, Yamei Luo[1]☯, Yuting Liu[2]☯, Yaru Luo[1], Liangce Wang[1], Shengsong Gou[3], Yuling Peng[1], Rui Wei[1], Da Jia[3], Yuan Wang[1], Shangbang Gao[2]*, Yan Zhang🄳[1]*

1 National Clinical Research Center for Geriatrics, State Key Laboratory of Biotherapy, West China Hospital, Sichuan University, Chengdu, Sichuan, China, 2 Key Laboratory of Molecular Biophysics of the Ministry of Education, College of Life Science and Technology, Huazhong University of Science and Technology, Wuhan, China, 3 Key Laboratory of Birth Defects and Related Diseases of Women and Children, Department of Pediatrics, State Key Laboratory of Biotherapy, West China Second University Hospital, Sichuan University, Chengdu, China

☯ These authors contributed equally to this work.
* sgao@hust.edu.cn (SG); yanzhang@scu.edu.cn (YZ)

**Data Availability Statement:** All relevant data are within the paper and its Supporting Information files.

## Abstract

Global increase of life expectancy is rarely accompanied by increased health span, calling for a greater understanding of age-associated behavioral decline. Motor independence is strongly associated with the quality of life of elderly people, yet the regulators for motor aging have not been systematically explored. Here, we designed a fast and efficient genome-wide screening assay in *Caenorhabditis elegans* and identified 34 consistent genes as potential regulators of motor aging. Among the top hits, we found VPS-34, the class III phosphatidylinositol 3-kinase that phosphorylates phosphatidylinositol (PI) to phosphatidylinositol 3-phosphate (PI(3)P), regulates motor function in aged but not young worms. It primarily functions in aged motor neurons by inhibiting PI(3)P-PI-PI(4)P conversion to reduce neurotransmission at the neuromuscular junction (NMJ). Genetic and pharmacological inhibition of VPS-34 improve neurotransmission and muscle integrity, ameliorating motor aging in both worms and mice. Thus, our genome-wide screening revealed an evolutionarily conserved, actionable target to delay motor aging and prolong health span.

Contrary to popular belief that life span and health span are strongly correlated, the global increase of life expectancy over the past decades is rarely accompanied by increased health span [1–4]. Since aging is characterized by functional decline of multiple organs and tissues, the key to healthy aging is to delay or rescue the decline of essential physiological functions. Motor independence is strongly associated with the quality of life of elderly people, yet motor aging is a common, conserved biological process from worms to humans, leading to frailty, loss of motor independence, falling, and even death [5,6]. To date, it is still challenging to

**Funding:** This work was supported by National Clinical Research Center for Geriatrics, West China Hospital, Sichuan University (Z2021JC006 to YZ), the National Natural Science Foundation of China (82173179 to YZ), Natural Science Foundation of Sichuan Province (2022NSFSC1402 to YZ), the Major International (Regional) Joint Research Project (32020103007 to SG) and the National Key Research and Development Program of China (2022YFA1206001 to SG). The funders had no role in study design, data collection and analysis, decision to publish, or preparation of the manuscript.

**Competing interests:** The authors have declared that no competing interests exist.

**Abbreviations:** KD, knockdown; mPSC, miniature postsynaptic current; NGM, nematode growth medium; NMJ, neuromuscular junction; RNAi, RNA interference; SDH, succinate dehydrogenase; VNC, ventral nerve cord.

identify evolutionarily conserved mechanisms that can be exploited to delay or ameliorate motor aging.

Nematode *Caenorhabditis elegans* (*C. elegans*) is a widely used animal model in aging studies. Like humans, *C. elegans* exhibits motor decline during aging, which mainly involves histological and functional deterioration of motor neurons and muscles [7–10]. Several studies have used candidate approaches to determine genes involved in the aging of motor neurons or muscles in both worm and mouse models [11–16]. However, the regulators for motor aging have not been systematically explored, partly due to a lack of high-throughput screening method.

In this study, we designed a fast and efficient genome-wide screening assay in *C. elegans* to systematically identify potential regulators of motor aging. Among the top hits, we functionally validated the role of VPS-34 in regulating motor aging and revealed its cell type-specific mechanisms. Combining genetics, pharmacology, and in situ electrophysiology, we further demonstrated that partial inhibition of VPS-34 significantly improved the neuromuscular synaptic transmission and the muscle integrity, which ameliorate motor aging in both worms and mice. Thus, our results revealed an evolutionarily conserved, actionable target to delay motor aging and prolong health span.

## Results

### Genome-wide RNAi screening in aged worms identifies candidate regulators of motor aging

To identify molecular regulators for motor aging, we first designed a simple assay to measure the motor activity of the worms as they age. We plated the worms in a 1 cm diameter circle in the center of the culture plate and quantified the percentage of worms that moved outside the circle (out-of-circle ratio) after 1 h (Fig 1A and Methods). To test if this assay could reflect the gradual decline of motor activity during aging, we monitored the worms from the young adult (day 1, D1) to aged (D9) stages and quantified the out-of-circle ratio every other day. Indeed, we observed a gradual reduction of the out-of-circle ratio from about 80% to 0%, and most of the worms lost their motor activity on D9 (S1A Fig). These observations are consistent with the previously reported timeline of locomotion decline as worms age [3,10]. Based on this assay, we performed genome-wide RNA interference (RNAi) screen on an RNAi hypersensitive VH624 strain [17] from larval stage L4 to identify the regulators of motor aging, using increased out-of-circle ratio on D9 as an indicator of improved motor activity. Through 3 rounds of screen, we found 34 consistent positive hits, whose knockdown consistently increased the out-of-circle ratios (S1B Fig and S1 Table). We then performed gene-network analysis of these hits using GeneMANIA [18] to reveal the interactions between their encoded proteins and partners (S1C Fig). Gene Ontology analysis reveals that these genes are enriched in pathways including phosphatidylinositol metabolic process, ligase activity, protein modification, cellular metal ion homeostasis (S2 Table). In line with the purpose of our screening, 5 positive hits are involved in aging or age-related neurodegeneration [19–24] (S1C Fig and S1 Table).

### VPS-34 regulates age-associated motor decline

Among our most prominent hits, we focused on *vps-34* (human homolog *PIK3C3*), a class III phosphatidylinositol 3-kinase responsible for the production of phosphatidylinositol 3-phosphate (PI(3)P), which plays important roles in endocytic sorting [25–28], engulfment [29], and autophagy [30,31], but has not been implicated in motor aging. We found that 65%

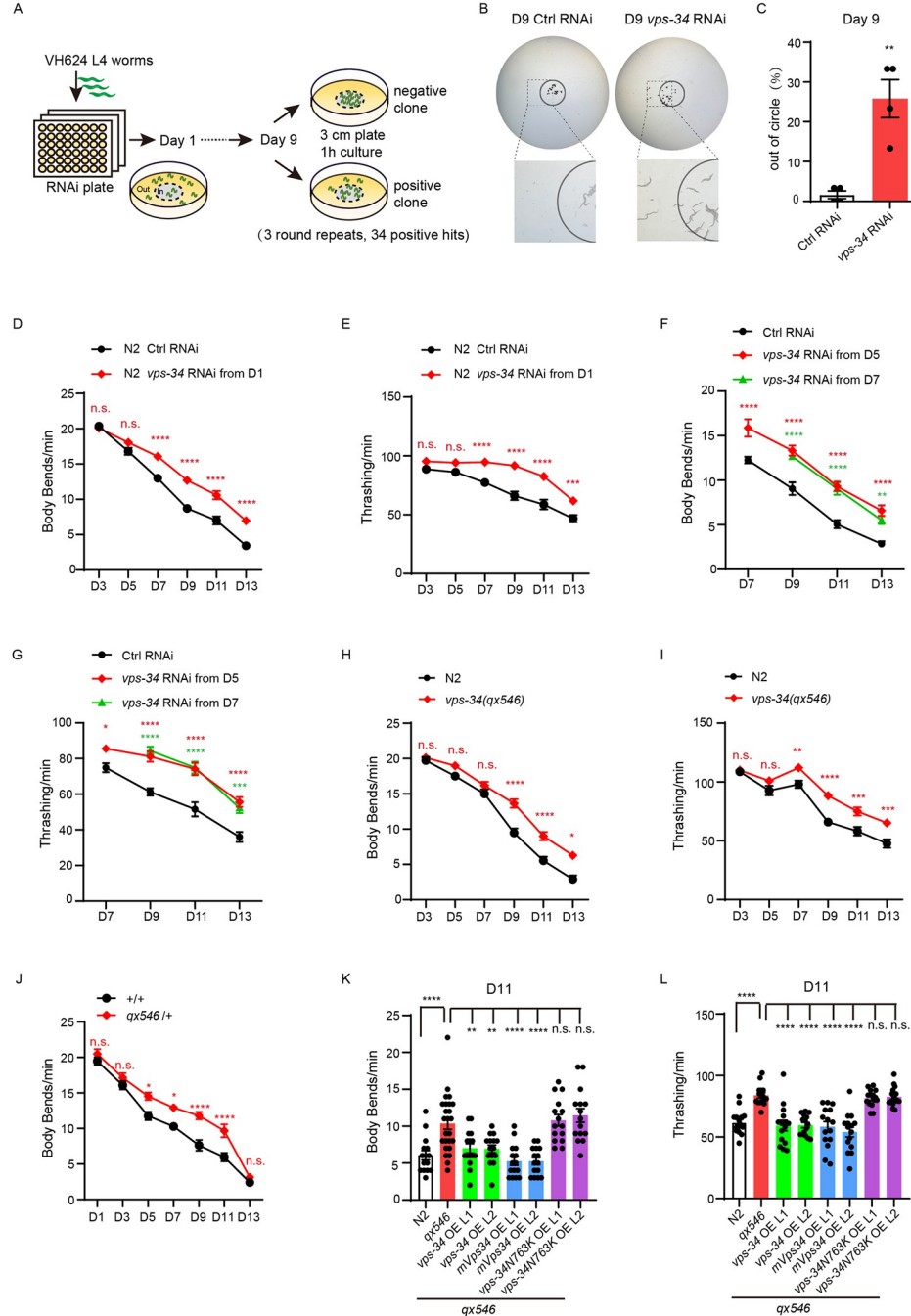

**Fig 1. Genome-wide screening in *C. elegans* identifies *vps-34* as the regulator of motor aging.** (**A**) Illustration showing the design of genome-wide RNAi screening for motor aging regulators. RNAi-fed worms exhibiting more than 10% increase of out-of-circle ratio at adulthood Day 9 (D9) were considered positive hits. Three rounds of screening were performed. (**B**) Representative images showing the distribution of control (ctrl) and *vps-34* RNAi worms inside and outside the circle at D9. (**C**) The percentages of out-of-circle worms in ctrl and *vps-34* RNAi groups are compared. *n* = 4 independent experiments; 30 worms were tested per group. (**D to I**) Motor behavior assays (Body bends, D, F, H; Thrashing, E, G, I) comparing ctrl and *vps-34* RNAi, N2 and *vps-34(qx546)* worms at multiple time points during aging. RNAi started from D1 (D and E) or D5 and D7 (F and G). (D and E, H and I) 50 worms per group. (F and G) 15 worms per group. *n* = 3 independent experiments. (**J**) Quantification of body bend frequency in heterozygous *vps-34(qx546)/+* and *+/+* worms at multiple time points during aging. Over 15 worms per group. *n* = 2 independent experiments. (**K and L**) Motor behavior assays (Body bends, K; Thrashing, L) comparing N2, *vps-34 (qx546)*, and *vps-34(qx546)* overexpressing (OE) worm *vps-34*, mouse *Vps34* (*mVps34*), or worm kinase-dead mutant *vps-34N763K* at D11. Two independent transgenic lines (L1 and L2) for each genotype were tested. Over 15 worms per

group. *n* = 3 independent experiments. Error bars, SEM. \*\*$P < 0.01$, \*\*\*$P < 0.001$, \*\*\*\*$P < 0.0001$. n.s., not significant. **C**, Unpaired two-tailed Student's *t* test; **D, E, H, I, J**, two-way ANOVA with Sidak's multiple comparisons test. **F, G**, Two-way ANOVA with Dunnett's multiple comparisons test. **K, L**, One-way ANOVA with Dunnett's multiple comparisons test. Raw data can be found in the Supporting information (S1 Data). RNAi, RNA interference.

knockdown (KD) of *vps-34* expression dramatically increased the out-of-circle ratio to about 25% on D9 (Figs 1B, 1C and S1D), suggesting that *vps-34* may be involved in regulating motor aging. We further used standard motor function assays (body bending and thrashing) to confirm this phenotype [32] (Methods). Consistently, in the VH624 strain *vps-34* RNAi from D1 improved motor activity throughout the aging process (S1E and S1F Fig). To test whether the KD phenotype is strain-specific, we examined the effect of *vps-34* RNAi in the commonly used wild-type N2 strain and found that *vps-34* RNAi also delayed motor function decline during aging (Fig 1D and 1E). To investigate whether *vps-34* KD from middle-aged worms could achieve similar effects, we started *vps-34* RNAi from D5 or D7 in both VH624 and N2 strains and found consistent improvement of motor function during aging through body bending and thrashing assays (Figs 1F, 1G, S1G and S1H). To test whether there is a key time point to inhibit VPS-34, we performed additional *vps-34* RNAi from D9 (old-age). While we did not observe a significant improvement on D11 and D13, at later time points (D15 and D17), *vps-34* RNAi improved the thrashing and/or body bend (S1I and S1J Fig). In contrast, D5 or D7 (mid-age) RNAi has a more immediate and long-lasting rescue effect. Thus, while VPS-34 inhibition from both mid-age and old-age are beneficial to delay motor aging, it appears that inhibition no later than mid-age has more dramatic effect. Together, these results demonstrate that *vps-34* KD could delay motor function decline during aging.

To test *vps-34* function in a more physiological setting, we sought to validate our results in *vps-34* mutant strains. *vps-34* null mutant causes developmental arrest at mid-stage larva [33], precluding its use for aging studies. Thus, we chose a recently developed *vps-34* mutant allele *qx546*. *qx546* carries mutations in 2 K63-linked ubiquitination sites, K348R and K352R, resulting in reduced level of VPS-34 protein [29]. We first confirmed the published results that VPS-34 protein in *qx546* worms is reduced to 50% of the wild type level (S2A Fig). Furthermore, the predicted protein structure of the mutant VPS-34 (K348RK352R) is similar to the wild type (S2B Fig), suggesting that the remaining VPS-34 protein in *qx546* worms may retain its normal function. *qx546* worms exhibit normal brood size and development (S2C–S2F Fig) and have a moderately increased median lifespan compared to wild-type worms (S2G Fig and S3 Table). Like *vps-34* RNAi, *vps-34(qx546)* shows an increased frequency of body bends (120% increase) and thrashing (80% increase) in aged worms but not in young worms (Fig 1H and 1I). Interestingly, we also observed improved motor activity in aged but not young *qx546* heterozygous worms (Fig 1J). Overexpressing VPS-34(K348RK352R) in wild-type worms did not increase the motor activity in aged worms (S2H and S2I Fig), arguing against its role as a gain-of-function mutant. These data demonstrate that *vps-34* is a haploinsufficient gene that regulates age-associated locomotion decline.

To ascertain that the improved motor activity in *vps-34(qx546)* mutant is dependent on reduced level of VPS-34, we overexpressed worm and mouse VPS-34 in *vps-34(qx546)* worms, each using 2 independent transgenic lines (L1 and L2). The motor function of these worms reduced to the level of controls in body bending and thrashing assays on D11 (Fig 1K and 1L), supporting an evolutionarily conserved role of VPS-34. To test whether VPS-34 functions through its kinase activity, we overexpressed kinase-dead VPS-34 in 2 transgenic lines [25] and observed no difference in motor activity in aged worms compared to *vps-34(qx546)* (Fig 1K and 1L), suggesting VPS-34 kinase activity is required for its regulation of motor aging.

We also examined the effects of VPS-34 partial inhibition on multiple other organs, including the epidermis, intestine, and pharynx. In the epidermis of aged worms, the number of vesicular lysosomes is reduced but the volume of each lysosome was increased compared to young worms, which is characteristic of age-related lysosome dysfunction [34]. These phenotypes are consistently rescued by *vps-34(qx546)* mutation and VPS-34 inhibitor SAR405 inhibition (S2J–S2O Fig). The intestinal atrophy is also a prominent aging feature that is closely related to the lifespan, characterized by reduced relative intestinal width [35]. *vps-34(qx546)* mutation increased the relative intestinal width on D9 (S2P Fig). Finally, we examined the pharyngeal pumping rate, which also declines with aging, but did not observe significant rescue effect of *qx546* mutation and SAR405 inhibition in aged worms (S2Q Fig). Thus, partial inhibition of VPS-34 appears to show protective effects on the intestine and epidermis, but not pharynx in aged *C. elegans*.

## VPS-34 regulates motor aging primarily through its expression in motor neurons

Next, we sought to determine in which cell type(s) VPS-34 primarily exerts its function to regulate motor aging. We used a transgenic reporter under the control of *vps-34* promoter to monitor the expression of *vps-34*. VPS-34 is expressed in multiple tissues including muscles, neurons, and intestine, with the highest expression in neurons (Fig 2A).

Since the locomotion of *C. elegans* was controlled by body wall muscles and neurons, we first examined the body wall muscle integrity in *vps-34(qx546)* during aging. Age-related body wall muscle deterioration is characterized by a gradual increase of mitochondria fragmentation [9,36,37]. Thus, we used the PD4251 strain (P*myo-3*::mitoGFP) that carries a reporter for mitochondria in body wall muscles. We categorized the mitochondria morphology into 3 patterns from tubular (intact) to fragmented (Fig 2B). In wild-type aged worms, most of the mitochondria in the body wall muscle are fragmented in sharp contrast to young worms, whereas in aged *qx546* worms the proportion of fragmented mitochondria is significantly reduced (Fig 2C). These results indicate that reduced VPS-34 level in *qx546* improves the muscle integrity in aged worms. To test whether this phenotype is regulated cell-autonomously or non-cell-autonomously, we overexpressed *vps-34* in muscles or neurons of *qx546* worms. We only observed an increase of fragmented mitochondria to the wild type level when *vps-34* is overexpressed in muscles (Fig 2C). Thus, *vps-34* cell-autonomously regulates age-associated deterioration of muscle integrity.

Next, we sought to dissect whether VPS34 regulates aging-associated motor behavioral decline through its function in muscles or neurons. We overexpressed *vps-34* under muscle (*myo-3*) or pan-neuron (*rgef-1*)-specific promoters in *vps-34(qx546)* worms. Interestingly, we found that neuron-specific, but not muscle-specific overexpression of *vps-34* reduced the motor activity to the control levels in aged worms (D9, D11) (Figs 2D, 2E, S3A and S3B), indicating that VPS-34 primarily functions in neurons to regulate motor aging.

To determine whether VPS-34 functions through specific neuronal subtypes, we similarly overexpressed *vps-34* using motor neuron (*acr-2+unc-25*) or sensory neuron (*osm-6*) specific promoters and found that only motor neuron-specific overexpression of *vps-34* reduced the motor activity to the control levels in aged *vps-34(qx546)* worms (Figs 2F, 2G, S3C and S3D). These data demonstrate that VPS-34 regulates motor aging primarily through its expression in motor neurons.

To determine whether VPS-34 regulates PI(3)P levels in neurons, we used a GFP reporter conjugated to the PI(3)P-binding domain FYVE to mark PI(3)P [29,33]. When expressed in neurons under *unc-119* promoter, the GFP::2×FYVE reporter exhibited a small puncta-like

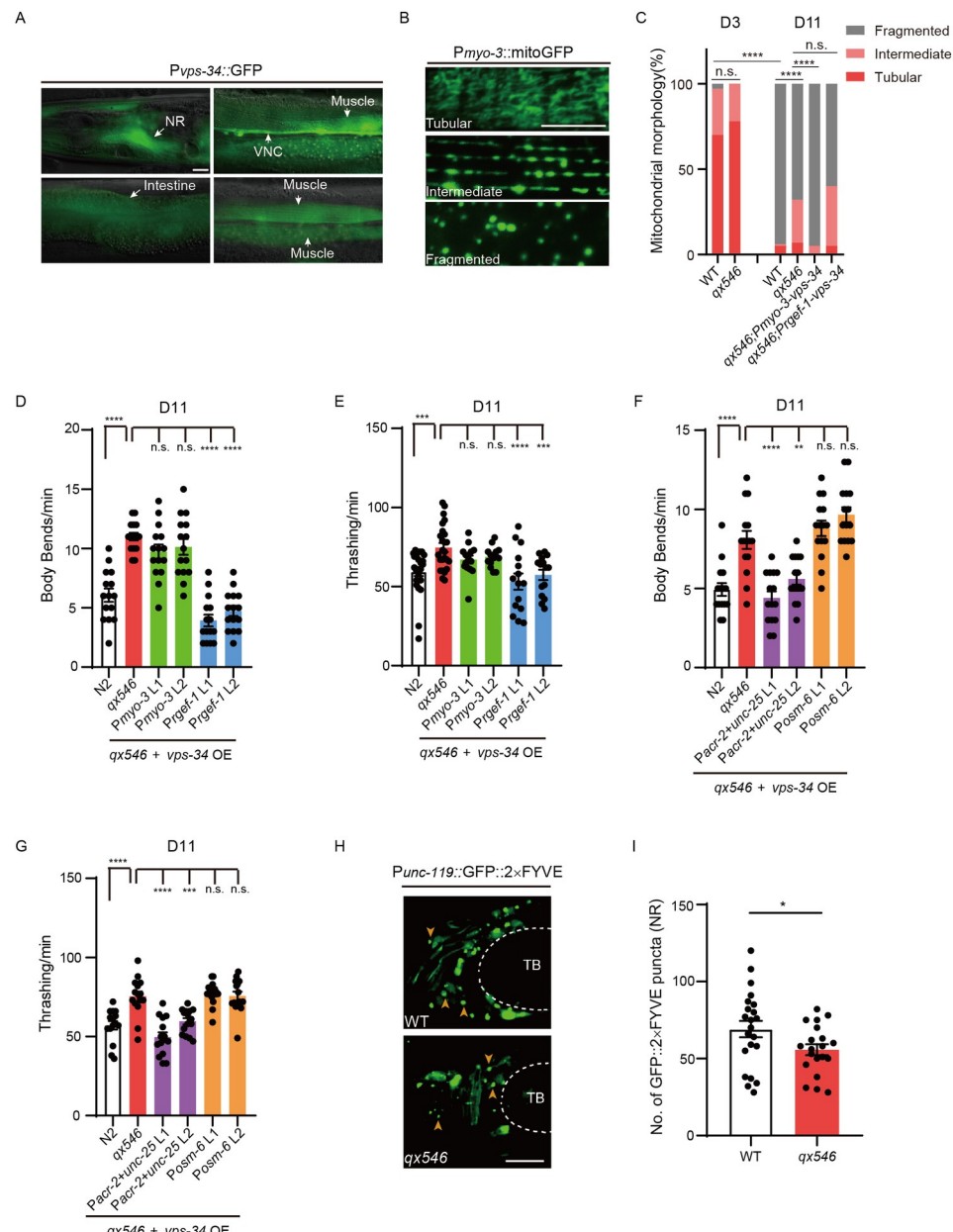

**Fig 2. VPS-34 primarily functions in motor neuron to regulate motor aging.** (**A**) Representative images showing *vps-34* transcriptional reporter P*vps-34*::GFP in multiple tissues of adult worms. NR: nerve ring. VNC: ventral nerve cord. (**B**) Representative images of 3 types of mitochondria morphology labeled by mitoGFP in the body wall muscles of P*myo-3*::mitoGFP worms. (**C**) Distribution of different types of mitochondria morphology in strains including WT, *qx546*, *qx546* overexpressing (OE) *vps-34* under body wall muscle-specific promoter (P*myo-3*) or pan-neuronal promoter (P*rgef-1*); 16–21 worms were imaged per genotype and approximately 300 muscle cells were assayed per genotype at each time point. *n* = 2 independent experiments. (**D** and **E**) Motor behavior assays (Body bends, D; Thrashing, E) comparing N2, *vps-34(qx546)*, and *vps-34(qx546)* overexpressing (OE) *vps-34* under pan-neuronal promoter (P*rgef-1*), body wall muscle-specific promoter (P*myo-3*) at D11. Two independent transgenic lines (L1 and L2) for each genotype were tested. Over 15 worms per group. *n* = 2 independent experiments. (**F** and **G**) Motor behavior assays (Body bends, F; Thrashing, G) comparing N2, *vps-34(qx546)*, and *vps-34(qx546)* overexpressing (OE) *vps-34* under motor neuron promoters (P*acr-2*+*unc-25*) or sensory neuron promoter (P*osm-6*) at D11. Two independent transgenic lines (L1 and L2) for each genotype were tested. Over 15 worms per group. *n* = 2 independent experiments. (**H**) Representative confocal images showing the expression of the PI(3)P reporter GFP::2×FYVE in the NR of wild type (WT) and *vps-34(qx546)* worms at D1. Orange arrowheads indicate PI(3)P puncta. TB, Terminal pharyngeal bulb. (**I**) Quantification of GFP::2×FYVE puncta numbers in the NR of WT and *vps-34(qx546)* at D1. NR afront of TB was counted. Error bars, SEM. \**P* < 0.05, \*\**P* < 0.01, \*\*\**P* < 0.001, \*\*\*\**P* < 0.0001. n.s., not significant.

**C**, Chi-square test. **D–G,** One-way ANOVA with Dunnett's multiple comparisons test. **I**, Unpaired two-tailed Student's *t* test. Scale bars, 10 μm. Raw data can be found in the Supporting information (S1 Data).

pattern (Figs 2H and S3E). The number of GFP::2×FYVE puncta in neurons is reduced (20% reduction) in *vps-34(qx546)* mutant compared to wild type (Fig 2H and 2I), indicating that VPS-34 regulates PI(3)P levels in neurons. To confirm the specificity of GFP::2×FYVE for PI(3)P, we used a mutant form of FYVE (C215S) that could not bind PI(3)P [38,39]. GFP::2×FYVE(C215S) expressed in neuron exhibited a smear-like pattern clearly distinguishable from GFP::2×FYVE (S3E Fig).

## VPS-34 negatively regulates synaptic transmission in aged worms

To further investigate the function of VPS-34 in neurons, we performed in situ electrophysiological recordings of synaptic transmission at neuromuscular junctions (NMJs) between motor neurons and body wall muscles (Methods). The frequency and amplitude of endogenous miniature postsynaptic currents (mPSC) were analyzed in both young and aged worms. The frequency reflects the number of released synaptic vehicles, while the amplitude correlates with both the neuronal activity and the number of muscle receptors. In wild-type N2 animals, we observed approximately 70% reduction of mPSC frequency on D11 compared to D3 (Fig 3A and 3B), which is consistent with a previous report [10]. Strikingly, *qx546* mutation dramatically enhanced the mPSC frequency on D11 to a level similar to D3 N2 worms (Fig 3A and 3B). This mPSC improvement was not observed on D3 animals (Fig 3A and 3B). Consistently, overexpression of *vps-34* in *qx546* motor neurons compromised the mPSC improvement in aged worms (Fig 3A and 3B). Next, we tested the effect of a highly potent and selective VPS34 inhibitor, SAR405 [40], on synaptic transmission in aged worms. SAR405 treatment on aged N2 worms, but not *vps-34(qx546)* worms, exhibited dramatically enhanced mPSC frequency on D11 (Fig 3A and 3B). Similarly, when we quantified the mPSC amplitude, we found similar results (Fig 3C). Notably, motor neuron-specific expression of *vps-34* could restore the enhanced amplitude in aged *qx546*, confirming a neuron—instead of muscle—dominant effect of VPS-34 in neurotransmission. Together, these data demonstrate that both genetic and pharmacological inhibition of *vps-34* could enhance the synaptic transmission in aged worms.

The electrophysiological data indicate VPS-34 could regulate the release of synaptic vesicles, consistent with its reported role in exocytosis [27]. Thus, we tested whether the key regulators of synaptic vesicle exocytosis, such as *unc-13*, *unc-18*, and *snt-1* [41–43], are downstream of VPS-34 inhibition. Indeed, KD of *unc-13*, *unc-18*, and *snt-1* in *vps-34* RNAi worms blocked motor activity improvement in aged worms (Fig 3D and 3E). Alternatively, VPS-34 might directly regulate the number of synapses. To test this possibility, we used GFP-tagged synaptic vesicle proteins RAB-3 [44,45] and SNB-1 [46] to quantify the synapse number in *vps-34* RNAi and *vps-34(qx546)* versus wild-type worms. We did not observe significant differences on D3 and D11 (S4A–S4H Fig), suggesting that inhibition of VPS-34 does not affect the synapse number in aged neurons. Together, these data support that VPS-34 primarily functions through its regulation of synaptic vesicle exocytosis.

## Inhibition of VPS-34 promotes PI(3)P-PI-PI(4)P conversion to improve motor function during aging

We next investigated the molecular mechanism underlying the regulation of exocytosis by VPS-34. It has been reported that the phosphoinositide conversion of PI(3)P-PI-PI(4)P promotes the recruitment of the exocyst tethering complex to enable membrane fusion and

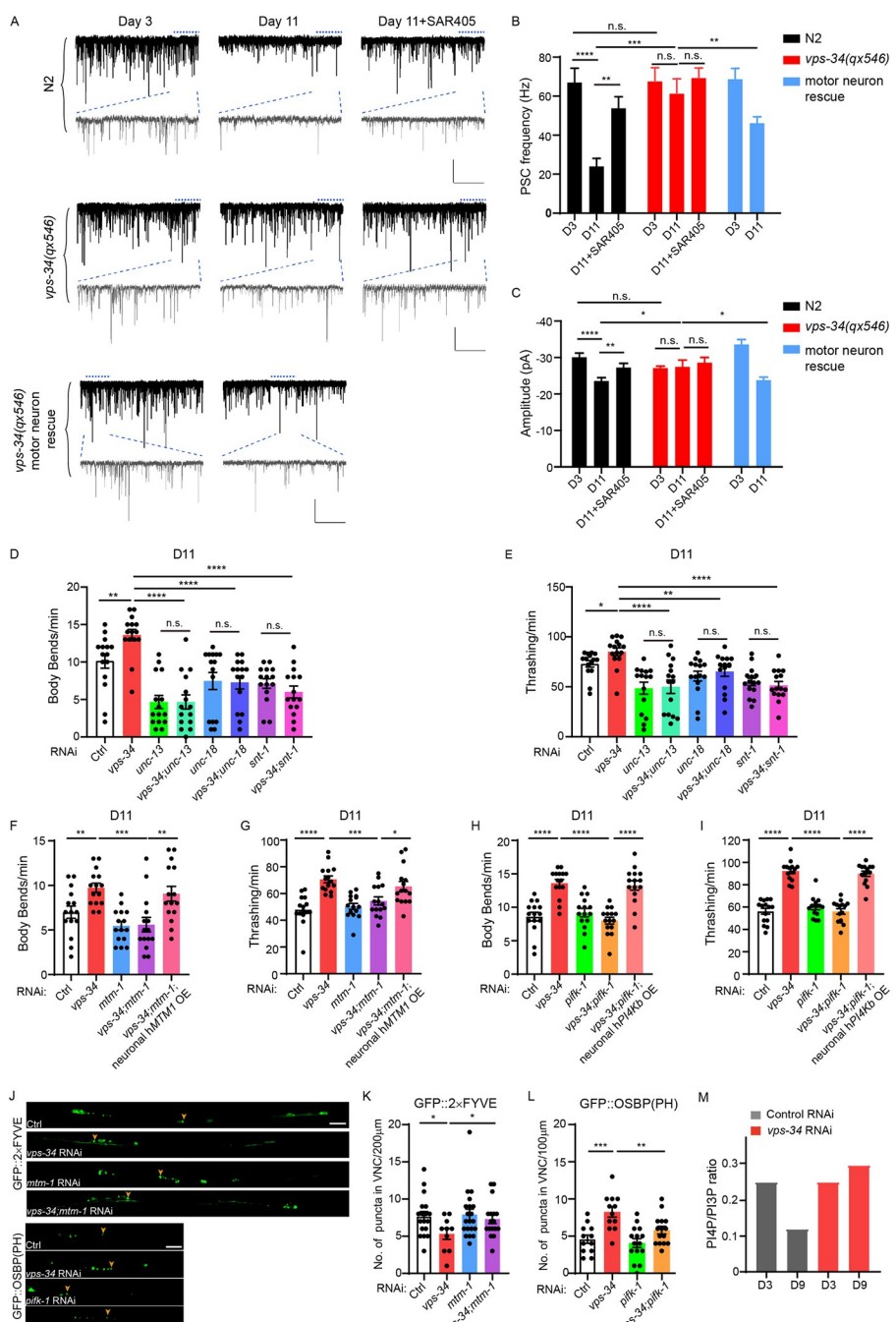

**Fig 3. Genetic and pharmacological inhibition of VPS-34 improve synaptic transmission in aged worms.** (**A**) Representative endogenous miniature PSC traces were recorded from NMJs at the VNC in N2, *vps-34(qx546)*, *vps-34 (qx546)*+motor neuron rescue groups at young (D3) or aged (D11) stages, with or without SAR405 treatment. The worms were treated with SAR405 from D1. Membrane voltage was clamped at −60 mV. Scale bar: 50 pA, 1.2 s (240 ms). (**B** and **C**) Summary of PSC frequency (B) and amplitude (C) for A. *n* = 10 worms each genotype or each treatment. (**D** and **E**) Motor behavior assays (Body bends, D; Thrashing, E) comparing ctrl RNAi, *vps-34* RNAi (mixed 1:1 with HT115 control RNAi), *unc-13* RNAi (mixed 1:1 with HT115 control RNAi), *vps-34; unc-13* double RNAi, *unc-18* RNAi (mixed 1:1 with HT115 control RNAi), *vps-34; unc-18* double RNAi, *snt-1* RNAi (mixed 1:1 with HT115 control RNAi), *vps-34; snt-1* double RNAi groups at D11. (**F** and **G**) Motor behavior assays (Body bends, F; Thrashing, G) comparing ctrl RNAi, *mtm-1* RNAi (mixed 1:1 with HT115 control RNAi), *vps-34* RNAi (mixed 1:1 with HT115 control RNAi), *vps-34; mtm-1* double RNAi, neuronal OE of human *MTM1* (h*MTM1*) in *vps-34; mtm-1* double RNAi groups at D11. (**H** and **I**) Motor behavior assays (Body bends, H; Thrashing, I) comparing *vps-34* RNAi (mixed 1:1 with HT115 control RNAi), *pifk-1* RNAi (mixed 1:1 with HT115 control RNAi), *vps-34; pifk-1* double RNAi, neuronal

OE of human *PI4Kb* (h*PI4Kb*) in *vps-34; pifk-1* double RNAi at D11. Fifteen worms per group. *n* = 2 independent experiments. Wild-type strain was used. (**J**) Representative fluorescent images of GFP::2×FYVE and GFP::OSBP(PH) in the VNC at D9. (**K**) Bar graph shows the number of GFP::2×FYVE positive puncta in ctrl RNAi, *vps-34* RNAi, *mtm-1* RNAi, *vps-34*; *mtm-1* double RNAi groups at D9; 10–20 worms per group. (**L**) Bar graph shows the number of GFP::OSBP(PH) positive puncta in ctrl RNAi, *pifk-1* RNAi, *vps-34* RNAi, *vps-34*; *pifk-1* double-RNAi groups at D9. Over 12 worms per group. *n* = 2 independent experiments. (**M**) Ratio of PI4P/PI3P puncta numbers in ctrl RNAi or *vps-34* RNAi at D3 and D9. Error bars, SEM. \**P* < 0.05, \*\**P* < 0.01, \*\*\**P* < 0.001, \*\*\*\**P* < 0.0001. n.s., not significant. **B–I, K, L**, Unpaired two-tailed Student's *t* test. Scale bars, 10 μm. Raw data can be found in the Supporting information (S1 Data). NMJ, neuromuscular junction; RNAi, RNA interference; VNC, ventral nerve cord.

facilitate exocytosis [27]. Since VPS-34 mediates the conversion of PI to PI(3)P, we tested whether this process is important for *vps-34* function in motor aging. We used RNAi to KD both *vps-34* and *mtm-1*, a PI(3)P phosphatase that plays an opposite role by mediating the conversion of PI(3)P to PI [27,47]. The double KD worms no longer exhibited increased motor activity, while expressing human *MTM-1* in neurons could rescue the phenotype (Fig 3F and 3G). In addition, we found that the number of GFP::2×FYVE labeled PI(3)P is restored in double KD worms on D9 compared to *vps-34* RNAi (Fig 3J and 3K). These data suggest that inhibition of VPS-34 improved motor function through increased PI(3)P-PI conversion.

To test whether reduced PI(3)P results in increased PI(4)P to promote exocytosis, we used RNAi to KD both *vps-34* and *pifk-1*, an orthologue of human PI4Kb that converts PI to PI(4)P in *C. elegans* [48,49]. We found that *pifk-1* RNAi could also block the motor improvement in aged *vps-34* RNAi worms, while expressing human *PI4Kb* in neurons could rescue the phenotype (Fig 3H and 3I). In addition, we quantified the PI(4)P level in neuron using a PI(4)P reporter strain GFP::OSBP(PH) [50] and found that *vps-34* RNAi results in a 2-fold increase of GFP::OSBP(PH) puncta in aged worms. In *pifk-1* and *vps-34* double KD worms, we observed reduced GFP::OSBP(PH) puncta compared to *vps-34* RNAi in aged worms (Fig 3J and 3L). Thus, the PI(4)P level is restored by *vps-34* inhibition, which is implicated in vesicle exocytosis [27]. To test whether this process is age dependent, we calculated the ratio of PI4P versus PI3P using reporter strains, and found that it does decline with aging in control worms, but could be rescued by *vps-34* RNAi (Fig 3M). Together, these data indicate that inhibition of VPS-34 promotes PI(3)P-PI-PI(4)P conversion to improve motor function during aging.

## VPS-34-regulated autophagy and endocytosis do not play a major role in motor aging

In addition to exocytosis, VPS-34 is also implicated in autophagy and endocytosis [26,28,31], which may play a role in motor aging. To test whether partial inhibition of VPS-34 impacts on autophagy in neurons, we examined the neuronal expression of a classic autophagy reporter GFP::LGG-1 and found that its level is gradually reduced in the nerve ring during aging (S5A and S5B Fig). *vps-34(qx546)* and *vps-34* RNAi worms showed a 30% to 40% reduction of GFP::LGG-1 in the nerve ring compared to wild type (S5C–S5F Fig). Thus, inhibition of VPS-34 reduces autophagy in neurons, consistent with previous reports [30]. We then investigated whether inhibition of autophagy could improve motor function in aged worms. When we RNAi knockdown the autophagy genes (*pha-4, atg-13, unc-51, bec-1, lgg-1, lgg-2, lgg-3, cup-5*), or used autophagy mutants, we did not observe significant increase of motor activity in aged worms (S5G–S5I Fig). Furthermore, we performed *vps-34* RNAi in autophagy mutants for genes essential for different autophagic stages, such as nucleation (*epg-8, atg-9*), elongation (*atg-3, atg-2, atg-9*), fusion (*epg-5*). Since some of the autophagy mutants exhibited apparently reduced basal level motility, we calculated the fold changes of *vps-34* RNAi versus control RNAi in each group and compared the rescue effect of *vps-34* RNAi in a wild-type or

autophagy mutant. We observe similar levels of rescue effects across different autophagy mutant background: wild-type (1.5-fold) and other autophagy mutant such as *epg-8* (1.5-fold), *atg-9* (1.5-fold), *atg-3* (1.5-fold), *epg-5* (1.8-fold), *atg-2* (1.3-fold) (S5J Fig). Together, these data indicate that VPS-34-regulated autophagy is not essential for motor aging.

We further examined the endocytic trafficking in the *vps-34(qx546)* neurons, since reduced PI3P level in *vps-34(qx546)* may affect the transition from early endosome to the late endosome [51]. We used 2 fluorescent reporter strains to label early and late endosome markers, RAB-5 and RAB-7 [52]. In the dorsal nerve cord, the puncta number and fluorescence intensity of YFP::RAB-5 were similar between *vps-34(qx546)* and wild type (S6A and S6B Fig). The puncta number of GFP::RAB-7 is also unchanged. Although the fluorescence intensity is mildly reduced in young worms, there is no change in aged *qx546* worms (S6C and S6D Fig). Thus, reduced VPS-34 level does not dramatically impact on the endocytosis process in neurons.

## Pharmacological inhibition of VPS-34 ameliorates the motor decline in aged worms and mice

To investigate whether VPS-34 is an evolutionarily conserved, actionable target for motor aging, we tested the VPS34 inhibitor SAR405 on both worms and mice. We first analyzed the effect of SAR405 treatment on PI(3)P production in the neurons of aged worms. GFP::2×FYVE worms treated with 20 μm SAR405 from D1 showed a 30% reduction of PI(3)P puncta in the ventral nerve cord (VNC) on D9 compared to those treated with vehicle (Fig 4A and 4B). We then treated wild-type worms with 20 μm SAR405 from young adult (D1) or middle age (D5 or D7), and found all the treatment strategies improved motor activity in aged worms, measured by body bending and thrashing assays (Fig 4C–4F). These data demonstrate that partial inhibition of VPS34 by SAR405 could ameliorate the motor decline during aging.

We then tested the effect of SAR405 in natural aging mice. We first confirmed that the expression of VPS34 colocalizes well with the neuronal marker NeuN [53] in the spinal cord (Fig 4G), which contains a large number of motor neurons. We then treated aged mice with VPS34 inhibitor SAR405 (5 mg/kg) by intraperitoneal injection for 1 month (Fig 4H and Methods). Compared to the vehicle-treated group, SAR405-treated mice did not exhibit weight loss (Fig 4I and 4J), but performed significantly better in the treadmill running assay (Fig 4K). Furthermore, histological staining shows that the cross-sectional area of muscle fibers is increased in SAR405-treated mice compared to controls, indicating SAR405 ameliorates muscle atrophy in aged mice (Fig 4L and 4M). SAR405 also improved the mitochondria function in the muscle of aged mice, evidenced by increased mitochondrial succinate dehydrogenase (SDH) staining, an indicator of mitochondrial activity [16,54] (Fig 4N and 4O). Together, these results show that SAR405 could enhance the motor activity and muscle performance in aged mice.

## Discussion

To delay or ameliorate motor aging could improve the quality of life for elderly people and promote their health span, yet the regulators for motor aging have not been systematically revealed, and there are few treatment options. In this study, we provide a global view of genes potentially regulating motor aging, and demonstrated that VPS34 is an evolutionarily conserved, actionable target to delay and ameliorate motor aging.

Previous studies have identified motor aging-associated regulators through candidate approaches, which act in either motor neurons/neuromuscular junctions or skeletal muscles [11–16]. To our knowledge, VPS34 is the first reported gene that simultaneously regulates

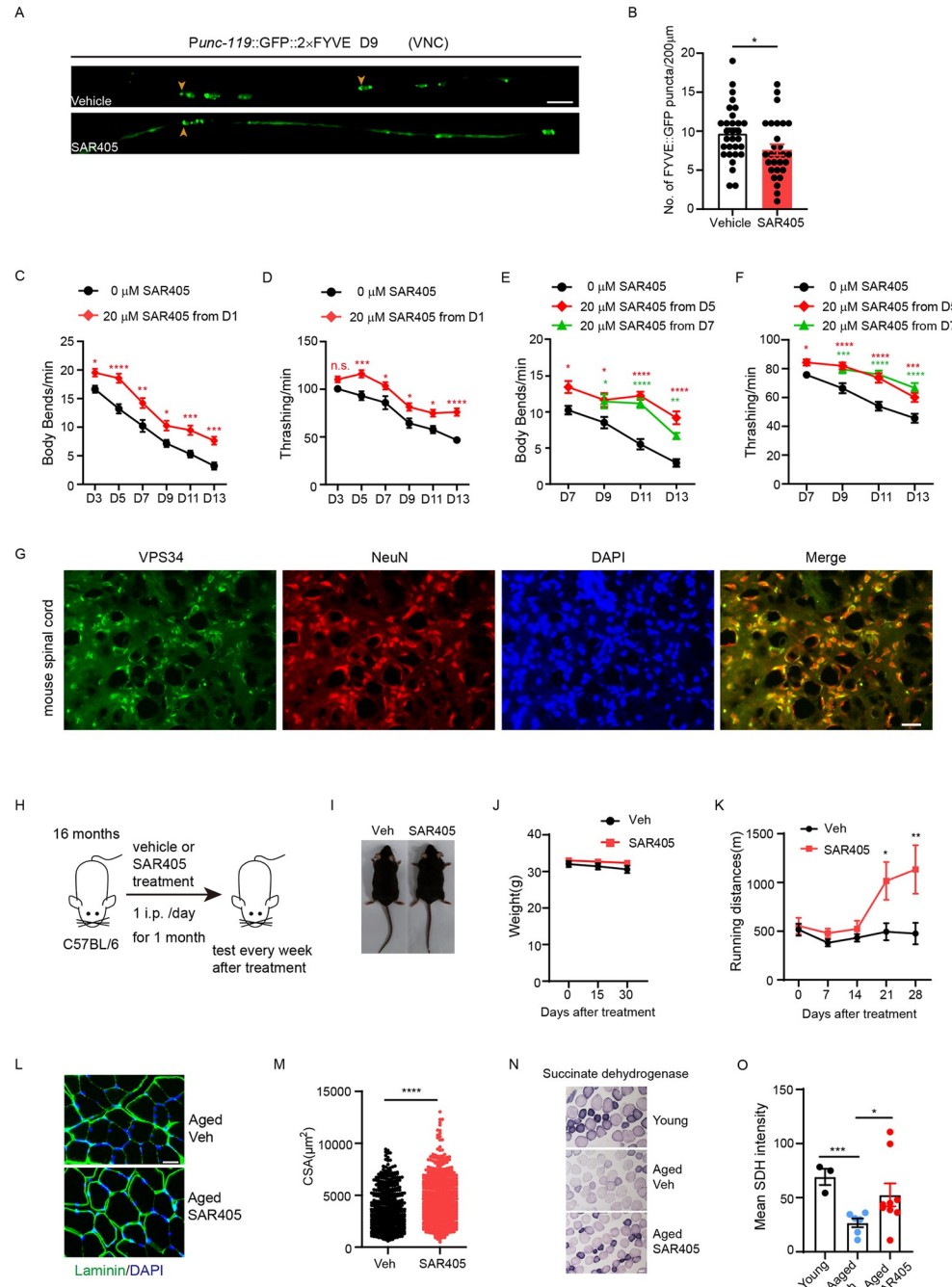

**Fig 4. Pharmacological inhibition of VPS34 improves motor activity in aged worms and mice.** (**A**) Representative confocal images of GFP::2×FYVE positive puncta in the VNC of vehicle or SAR405 (20 μm) treated worms at D9. Arrowheads indicate GFP:2×FYVE positive PI(3)P puncta. Scale bar, 10 μm. (**B**) Quantification of GFP::2xFYVE positive puncta in vehicle or SAR405 treated worms. Thirty worms per group. (**C** to **F**) Motor behavior assays (Body bends, C and E; Thrashing, D and F) comparing vehicle or SAR405 treated worms at multiple time points during aging. Worms were treated with SAR405 from D1 (C and D) or D5 and D7 (E and F). (C and D) Forty worms per group. (E and F) Fifteen worms per group. $n$ = 3 independent experiments. (**G**) Representative immunofluorescent staining of VPS34 (green) colocalizing with the neuronal marker NeuN (red) in mouse spinal cord sections. DAPI (blue) labels nuclei. Scale bar: 50 μm. (**H**) The scheme of SAR405 treatment on aged mice. (**I**) Representative whole-mount images of aged mice after treatment. (**J**) The body weight of mice was measured at day 15 and day 30 after treatment started. (**K**) Running distance to exhaustion on treadmill for vehicle or SAR405-treated aged mice after different treatment periods. Vehicle group (Veh), $n$ = 7. SAR405 group, $n$ = 9. (**L**) Representative staining images of Laminin (green) and DAPI (blue) on *Tibialis anterior* (TA) muscle cross-sections. Scale bar: 50 μm. (**M**) Average area

of TA myofiber CSA in vehicle or SAR405 treated aged mice. Vehicle group (Veh), 473 myofiber from 7 mice. SAR405 group, 607 myofiber from 9 mice. (**N**) Representative images of SDH staining on TA muscle from young and vehicle or SAR405-treated aged mice. Scale bar: 50 μm. (**O**) Mean SDH intensity was quantified in TA myofiber. Young, $n = 3$. Veh, $n = 6$. SAR405, $n = 9$. Error bars, SEM. $^*P < 0.05$, $^{**}P < 0.01$, $^{***}P < 0.001$, $^{****}P < 0.0001$. n.s., not significant. No label, $P > 0.05$. **C, D, J, K,** Two-way ANOVA with Sidak's multiple comparisons test. **B, M, O,** Unpaired two-tailed Student's *t* test. **E, F,** Two-way ANOVA with Dunnett's multiple comparisons test. Raw data can be found in the Supporting information (S1 Data). SDH, succinate dehydrogenase; VNC, ventral nerve cord.

neurotransmission of motor neurons and muscle integrity during aging, likely through cell type-specific mechanisms. Thus, it is a promising target that can be exploited to improve both aged neurons and muscle, as demonstrated by our SAR405 treatment experiments. Although the motor function improvement in our behavioral assays can be mainly attributed to the improved neurotransmission of motor neurons, additional assays to specifically test the functional improvement of muscle after VPS34 inhibition could further delineate the benefits of our treatment strategy.

We further pinpointed the molecular mechanism underlying improved motor neuron function after VPS34 inhibition. Through multiple genetic experiments, we showed that autophagy and endocytosis, 2 processes that involves VPS34, unlikely plays a major role in VPS34-regulated motor aging. Rather, inhibition of VPS34 shifts the balance of the PI(3)P-PI-PI(4)P conversion, resulting in higher levels of PI(4)P essential for exocytosis [27], which consequently increased synaptic vesicle release.

It is worth noting that partial inhibition but not complete blockade of VPS34 is sufficient to delay or ameliorate motor aging. Previous studies have shown that germline knockout of *vps-34* in nematodes and mice cause developmental arrest [33,55]. Conditional knockout of Vps34 in mice causes degeneration and failure of organs [30,56–59]. Since VPS34 is the only class III PI3K regulating the PI(3)P level, it is conceivable that it may be required in multiple organs or cell types. We carefully examined the worms from *vps-34* RNAi, *qx546*, and SAR405 partial inhibition groups, and did not observe deleterious effects in terms of brood size, overall development, and lifespan. Thus, our observation that partial inhibition of VPS34 delays or ameliorates neuronal and muscular aging is not contradictory to previous studies, but rather offers a viable treatment strategy to promote healthy aging.

## Materials and methods

### Ethics statement

All animal studies were approved by the Animal Care and Use Committee of West China Hospital, Sichuan University. Animal ethical license number is 2021617A.

### *C. elegans* strains

*C. elegans* strains were cultured on standard nematode growth medium (NGM) plates seeded with *Escherichia coli* OP50 or HT115 bacteria at 20°C. Bristol N2 strain was used as the wild-type strain. Transgenic worms were generated by standard microinjection methods by injecting 40 ng/μl plasmid DNA. We obtained *vps-34(qx546)* from X. Wang (Institute of Biophysics, Chinese Academy of Sciences, Beijing, China), autophagy mutants from H. Zhang (Institute of Biophysics, Chinese Academy of Sciences, Beijing, China). TXJ1245 was kindly gifted by X. Tong (University of Science and Technology Shanghai, Shanghai, China). The mutants and reporter strains are listed in S4 Table.

## RNAi screening and RNAi assays

Over 12,000 RNAi clones from *C. elegans* Vidal RNAi library (Horizon Discovery) were cultured in liquid LB containing ampicillin (0.1 mg/ml) overnight and seeded on NGM culture plate containing 0.1 mg/ml ampicillin and 1 mM IPTG.

Synchronized L4 VH624 worms were washed from OP50 in M9 buffer, and 40 to 50 worms were placed on HT115 RNAi dishes. Bacteria containing the empty vector (L4440) was used as an RNAi control. FUDR (0.0125 mg/ml, Alfa Aesar, Cat# L16497) was added to prevent progeny production. Worms were cultured at 20˚C for 9 days and washed off the RNAi plates with M9 buffer. For each RNAi or control group, about 30 worms were placed at the center of a 1 cm circle on the plate for 1 h at 20˚C. We then quantified the ratio of worms that moved out of circle. An RNAi group exhibiting an out-of-circle ratio of >10% is considered a positive hit, which is further validated through 2 additional rounds of screening.

RNAi assays were performed using standard RNAi feeding method [60]. RNAi treatments started from day 1, unless mentioned otherwise in the figure legends or main text. For groups without FUDR treatment, worms were transferred daily to exclude the interference from offspring. Double RNAi targeting 2 genes were conducted by seeding 2 RNAi bacteria at a ratio of 1:1 on the NGM plates. The controls in the double RNAi are target gene RNAi mixed 1:1 with HT115 control bacteria.

## RT-qPCR

Synchronized worms were lysed in TRIzol reagent (Ambion, Cat#15596018). Total mRNA was extracted, reverse transcribe to cDNA using FastKing RT Kit with gDNase (TIANGEN, KR116-02). RT-qPCR was performed using iTaq Universal SYBR Green super mix (BioRad Cat#1725124) on BioRad CFX96 Real-Time PCR system. The expression level of the housekeeping gene *act-1*, *tba-1*, *ama-1*, *cdc-42* is used as an internal control for normalization.

Primer pairs used for the qPCR:

*act-1* Forward: ATGTGTGACGACGAGGTT;

*act-1* Reverse: GAAGCACTTGCGGTGAAC;

*ama-1* Forward: TTCTACTGCGGACGGCTTCTCA;

*ama-1* Reverse: CGACACGGCGGTATGATGGTTG;

*cdc-42* Forward: ATGCTCAGCGTTGACGCAGAAG;

*cdc-42* Reverse: TCCTGTTGTGGTGGGTCGAGAG;

*tba-1* Forward: CGCATCATCTCGCAGGTTGTGT;

*tba-1* Reverse: GTGGAGTGTACGCAGCCAATGG;

*vps-34* Forward: GCATCACGACGTATTGAAATGA;

*vps-34* Reverse: CCGAAACAATCCCAACACCA.

## Plasmid construction

Expression vectors used for generating transgenic strains were constructed using standard protocols and are listed in S5 Table. The primers used for plasmid construction are listed in S6 Table.

## Western blots

Wild-type and *qx546* worms (approximately 2,000 worms per group) were lysed by RIPA lysis buffer containing protease inhibitors. Protein samples were analyzed by SDS-PAGE gel, followed by the standard western blot procedures. The primary antibodies used were anti-VPS34 (Proteintech, Cat#12452-1-AP) and anti-H3 (Abcam, Cat#ab8580).

## Protein structure prediction

AlphaFold2 (https://colab.research.google.com/github/sokrypton/ColabFold/) [61] were used to predict protein structures of VPS-34 (WT) and VPS-34 (K348RK352R), visualized in Pymol v2.3.0 (https://github.com/schrodinger/pymol-open-source).

## Standard motor behavioral assays for *C. elegans*

Frequency of body bends: The body bends test was performed as previously described before [62]. Thin lawn of OP50 NGM dishes were prepared for recording the frequency of body bends in 1 min. All the body bends tests are blind scoring and repeat 2 to 3 times.

Frequency of thrashings: The thrashing assay was performed as previously described before [63]. Individual worm from different ages was counted for 1 min in 1 drop of M9 buffer. All the thrashing tests are blind scoring and repeat 2 to 3 times.

## *C. elegans* lifespan assay

Lifespan assay was performed at 20˚C. For each group, a total of approximately 120 L4 worms were placed on NGM plate and transferred daily. The first day of adulthood is calculated as day 1. Worms with protruding vulvae, crawled off plate, or exploded were excluded. Worms were counted every day and the ones that do not respond to gentle touch were scored dead.

## *C. elegans* brood size measurement

Individual late L4 worms were placed onto NGM plates seeded with *E. coli* OP50 and transferred to a fresh dish every 24 h until they do not produce new progeny. The number of eggs laid by each worm was counted every day. For each group, a total of 10 worms were counted. The hatching progeny were counted 2 to 3 days after eggs were laid. We also measured the time required for hatching progeny to grow to L4.

## Quantification of lysosome morphology

The number of NUC-1::CHERRY-positive vesicular lysosomes per unit area ($15 \times 15$ μm$^2$) was quantified. At least 5 puncta diameter of each animal were measured. Over 25 animals were quantified in each strain at each day.

## Pharyngeal pumping assay

Worms were placed on OP50-seeded plates and allowed to adapt for 10 min at room temperature (approximately 20˚C), and then, the pharyngeal pumping rate of approximately 15 animals were counted under a dissection microscope.

## Intestinal atrophy measurements

Approximately 15 worms were placed on agar pad containing 1 mM levamisole, and the intestinal atrophy of nematodes was observed under the DIC of microscope. For intestinal atrophy assay, we measured relative intestinal width behind the pharynx that is calculated

by intestinal width subtracting the luminal width and dividing by the body width as previously described [35].

## Microscopy and imaging analysis

Confocal images of fluorescently tagged fusion proteins were performed using an Olympus FV3000 under a 60× oil objective. Images were analyzed in ImageJ.

Worms were anaesthetized with 1 mM levamisole and imaged immediately. DIC and fluorescence images were captured by Olympus BX63 automatic fluorescence microscope and Hamamatsu sCMOS camera ORCA-Flash4.0. Images were processed and viewed by Olypums cellSens Dimension software.

## PI3P and PI4P puncta analysis

PI3P was recognized as condensed GFP::2×FYVE puncta with diameter between 0.2 μm and 1.8 μm. PI4P was recognized as condensed GFP::OSBP(PH) puncta with diameter between 0.2 μm and 1.1 μm. The region of interest is in the nerve ring (between first pharyngeal bulb and terminal pharyngeal bulb) and the VNC (vulva to tail part) of nematode.

## *C. elegans* muscle mitochondrial analysis

PD4251 worms were imaged at the microscope under a 100× objective. The mitochondria were divided into 3 types: the tubular mitochondria, the intermediate mitochondria (partial loss of tubular morphology), and the fragmented mitochondria (completely broken without tubular morphology). The number for each type of mitochondria in muscle cells was counted. The composition of mitochondria types in each experimental group was calculated and compared by GraphPad Prism8, using Chi-square test to determine statistical significance.

## Electrophysiology

For electrophysiological experiments, the nematode dissection method was based on the method of our previous study [64], and the hermaphroditic adult worms to be observed were glued to a coverslip covered with bathing solution. The integrity of the NMJ preparation was examined visually by DIC microscopy and the anterior muscle cells were patched using polished 4 to 6 MΩ resistant borosilicate pipettes. Membrane currents were recorded in a whole-cell configuration using a HEKA EPC-9 membrane clamp amplifier using PULSE software and processed with Igor Pro and Clampfit. Currents were recorded at a holding potential of –60 mV. Data were digitized at 10 kHz and filtered at 2.6 kHz. The pipette solution contains (in mM): K-gluconate 115; KCl 25; $CaCl_2$ 0.1; $MgCl_2$ 5; BAPTA 1; HEPES 10; $Na_2ATP$ 5; $Na_2GTP$ 0.5; cAMP 0.5; cGMP 0.5, pH 7.2 with KOH, approximately 320 mOsm. cAMP and cGMP were included to maintain the activity and longevity of the preparation. The bath solution consists of (in mM): NaCl 150; KCl 5; $CaCl_2$ 5; $MgCl_2$ 1; glucose 10; sucrose 5; HEPES 15, pH 7.3 with NaOH, approximately 330 mOsm. Chemicals and blockers were obtained from Sigma unless stated otherwise. Experiments were performed at room temperature (20 to 22˚C).

## VPS34 inhibitor SAR405 treatment

SAR405 (Selleck, Cat#s7682) was dissolved in DMSO at a stock concentration of 20 mg/ml. For treatment on *C. elegans*, synchronized worms at day 1, day 5, or day 7 were then placed in NGM plates with bacteria lawn containing 20 μm SAR405 or DMSO on top of the plates. Motor behavioral assays were measured 2 days later. For treatment on mice, aged (16 months)

wild-type C57BL/6 mice were treated daily with SAR405 (5 mg/kg) or vehicle (7.5% DMSO) through intraperitoneal injection.

## Mice and the treadmill test

Wild-type C57BL/6 male mice at 16 months of age were purchased from Beijing Vital River Laboratory Animal Technology (Beijing, China). Mice were housed in pressurized, individually ventilated cages (PIV/IVC) and maintained under specific-pathogen-free conditions, with free access to food and water in a 12 h light/dark cycle.

Mice treated with SAR405 or vehicle for 1 month were subjected to the treadmill test to measure and compare their motor activity. The treadmill was with initially set to 10 m/min for 10 min, followed by 15 m/min for 10 min, and finally adjusted to 20 m/min until the mice exhausted. The electric shock intensity was 1.5 mA, and the treadmill instrument (SANS, SA101) automatically recorded the total distance the mice moved on the treadmill during 100 shocks. Prior to testing, the mice were trained 4 times using the same procedure. Each mouse was measured 3 times.

## Immunofluorescence staining

Mice were anesthetized and perfused with PBS, followed by 4% paraformaldehyde. After perfusion, the spinal cord and tibialis anterior muscle were removed and fixed in 4% paraformaldehyde for 24 h, then transferred to 30% sucrose for 2 to 3 days. Dehydrated tissues were then embedded in O.C.T. compound (Tissue-Tek), frozen on dry ice, and sectioned at 10 μm thickness. The sections were stored at −80°C until use.

For immunofluorescence staining, frozen sections were dried at 42°C for 30 min, rinsed and rehydrated with PBS, and treated with 0.2% Triton X-100 in PBS for 20 min at RT. Sections were then blocked with 2% goat serum in PBS for 1 h at RT and incubated with primary antibodies overnight at 4°C. Primary antibodies were visualized by species-specific goat secondary antibodies conjugated to Alexa Fluor dyes (Alexa 488/555, Cat#R37118 and Cat#A-31570, 1:1,000, Invitrogen). Sections were then stained with DAPI (1 μg/mL) (Helixgen, Cat# HNFD-02) for 5 min. Slides were coverslipped and imaged under an Olympus BX51 fluorescent microscope. Primary antibodies used in this study were: VPS34 (Sangon biotech, Cat#D261326, 1:200), NeuN (Abcam, Cat#ab104224, 1:200), Laminin (Abcam, Cat#ab11575, 1:200). The average cross-section area of each myoblast and the percentage of myoblasts of different areas among all cells were measured by ImageJ.

## Muscle succinate dehydrogenase staining

Muscle mitochondrial activity was measured by SDH staining (tetrazolium salt method). Unfixed tibialis anterior muscle was cut into frozen sections at 10 μm thickness, stained at 37°C for 20 min using SDH staining kit (Solarbio, Cat#G2000), washed in deionized water, and then, imaged under the microscope. Mean SDH intensity was measured by ImageJ.

## Statistics

Experimental data were analyzed using GraphPad Prism (v.8). For all graphs, the error bars indicate the mean ± SEM. To compare data from different groups, unpaired two-tailed Student's $t$ test, two-way ANOVA with Sidak's multiple comparisons test, two-way ANOVA with Dunnett's multiple comparisons test, one-way ANOVA with Dunnett's multiple comparisons test, Chi-square test, or Kaplan–Meier method followed by the log-rank test were used as

indicated in the figure legends. Differences were considered significant if $P < 0.05$. Statistical significance is indicated by asterisks; $^*P < 0.05$; $^{**}P < 0.01$; $^{***}P < 0.001$; $^{****}P < 0.0001$.

## Supporting information

**S1 Fig. Knocking down *vps-34* improved the motor activity in aged, but not young VH624 worms.** (**A**) The percentage of VH624 worms that stayed inside or moved outside the circle at multiple time points during aging; 30–40 worms per group. $n = 3$ independent experiments. (**B**) Summarized D9 out-of-circle ratios for ctrl and 34 positive hits. *eat-2* RNAi was used as the positive control. (**C**) Co-expression network of identified positive genes and their partners. The network was constructed by GeneMANIA. Black and gray dots show the positive hits and their partners. Orange dots show the positive hits with related function in aging study. (**D**) qPCR comparing the transcription of *vps-34* in *vps-34* and control RNAi worms. The normalization control genes were used as *act-1*, *tba-1*, *ama-1*, *cdc-42*. (**E–H**) Motor behavior assays (Body bends, E and G; Thrashing, F and H) comparing ctrl and *vps-34* RNAi using VH624 worms at multiple time points during aging. RNAi started from D1 (E and F) or D5 and D7 (G and H). Fifteen worms per group. $n = 3$ independent experiments. (**I and J**) Relative motor functions (Body bends, I; Thrashing, J) comparing ctrl and *vps-34* RNAi using N2 worms at multiple time points during aging. Control RNAi D11 was set as 100%. RNAi started from D9. Fifty worms per group from D11 to D15, while the group of D17 scored over 20 animals. Error bars, SEM. $^*P < 0.05$, $^{**}P < 0.01$, $^{***}P < 0.001$, $^{****}P < 0.0001$. n.s., not significant. **B, D,** Unpaired two-tailed Student's *t* test. **E, F, I, J,** Two-way ANOVA with Sidak's multiple comparisons test. **G, H,** Two-way ANOVA with Dunnett's multiple comparisons test. Raw data can be found in the Supporting information (S1 Data).
(TIF)

**S2 Fig. *vps-34(qx546)* exhibits normal fertility, development, and moderately extended lifespan.** (A) Western blot shows the protein levels of VPS-34 in wild-type (N2) and *qx546* worms. (**B**) Protein structures of wild-type (WT) and mutant (*qx546*, K348RK352R) predicted by AlphaFold2 using root-mean-square-deviation (RMSD) of 0.412 Å between 519 pairs of atoms. Orange: full-length WT VPS-34; Cyan: 281–901 of VPS-34 K348RK352R mutant, which overlaps well with wild-type; Green: WT K348 and K352 sites; Purple: Mutant R348 R352 sites. (**C**) Quantification of progeny numbers at D1, D2, D3 and D4 of adulthood in N2 and *qx546* worms. $n = 10$ worms per genotype. (**D**) Total brood size in N2 and *qx546*. $n = 10$ worms per genotype. (**E**) The hatching ratio of eggs in wild-type (N2) and *vps-34(qx546)* worms. Progeny from 10 worms were calculated per genotype. (**F**) The proportion of N2 and *qx546* worms that developed to L4 stage at different time points, starting from 60 h after hatching. Progeny from 10 worms were calculated per genotype. (C–F) $n = 2$ independent experiments. (**G**) Survival analysis of N2 and *vps-34(qx546)*. $n = 3$ independent experiments. (**H and I**) Motor behavior assays (Body bends, H; Thrashing, I) comparing ctrl and VPS-34 K348RK352R OE in N2 worms at D11. Two transgenic lines (L1, L2) were tested. Fifteen worms per group. $n = 2$ independent experiments. (**J**) Representative images showing the lysosomal reporter NUC-1::mCherry in the epidermis of WT and *vps-34(qx546)* worms at D1, D9, and D13. Scale bar, 10 μm. (**K**) Quantification of NUC-1::mCherry puncta numbers in the epidermis of WT and v*ps-34(qx546)* at D1, D9, and D13. Over 25 worms per group. (**L**) Quantification of NUC-1::mCherry puncta diameter in the epidermis of WT and *vps-34(qx546)* at D1, D9, and D13. Over 25 worms per group. (**M**) Representative images showing the lysosomal reporter NUC-1::mCherry in the epidermis of vehicle or SAR405-treated worms at D9. Scale bar, 10 μm. (**N and O**) Quantification of NUC-1::mCherry puncta numbers (N) and diameter (O) in the epidermis of vehicle or SAR405-treated worms at D9. Over 25 worms per group. (**P**)

Relative intestinal width analysis in N2, *vps-34(qx546)* at D9. Over 20 worms per group. (**Q**) Pumping rate counting in N2, *vps-34(qx546)*, vehicle or SAR405-treated N2 worms at D11. Over 15 worms per group. Error bars, SEM. $*P < 0.05$, $**P < 0.01$, $***P < 0.001$, $****P < 0.0001$. All other points as $P > 0.05$. **C, F**, Two-way ANOVA with Sidak's multiple comparisons test. **G**, Log-rank (Mantel–Cox) test. **D, E, H, I, K, L, N-Q**, Unpaired two-tailed Student's *t* test. Raw data can be found in the Supporting information (S1 Data and S1 Raw Images).
(TIF)

**S3 Fig. *vps-34* primarily acts in motor neuron to regulate motor activity in aged worms.** (**A and B**) Motor behavior assays (Body bends, A; Thrashing, B) comparing N2, *qx546*, and *qx546* overexpressing (OE) *vps-34* under *vps-34* promoter (P*vps-34*), pan-neuronal promoter (P*rgef-1*), body wall muscle specific promoter (P*myo-3*) at D9. Two independent transgenic lines (L1 and L2) for each genotype were tested. Fifteen worms per group. $n = 2$ independent experiments. (**C and D**) Motor behavior assays (Body bends, C; Thrashing, D) comparing N2, *qx546*, and *qx546* overexpressing (OE) *vps-34* under motor neuron promoters (P*acr-2*+*unc-25*) or sensory neuron promoter (P*osm-6*) at D9. Two independent transgenic lines (L1 and L2) for each genotype were tested. Fifteen worms per group. $n = 2$ independent experiments. (**E**) Representative images of GFP::2xFYVE and GFP::2xFYVE (C215S) are shown in the nerve ring. Orange arrowheads indicate GFP[+] PI(3)P puncta. Scale bar, 10 μm. Error bars, SEM. $**P < 0.01$, $***P < 0.001$, $****P < 0.0001$. n.s., not significant. **A–D**, One-way ANOVA with Dunnett's multiple comparisons test. Raw data can be found in the Supporting information (S1 Data).
(TIF)

**S4 Fig. Partial inhibition of *vps-34* does not impact on the number of synapses.** (**A**) Representative fluorescent images of GFP::RAB-3 in the dorsal nerve cord axons from WT or *qx546* worms at D3 and D11. (**B**) Quantification of GFP::RAB-3 labeled synaptic puncta in (A). (**C**) Representative fluorescent images of GFP::RAB-3 under *unc-129* promoter in the dorsal nerve cord axons of excitatory neurons from ctrl or *vps-34* RNAi worms at D3 and D11. (**D**) Quantification of GFP::RAB-3 labeled synaptic puncta in (C). (**E**) Representative fluorescent images of SNB-1::GFP under *unc-25* promoter in the dorsal nerve cord axons of inhibitory neurons from WT or *qx546* worms at D1 and D11. (**F**) Quantification of SNB-1::GFP labeled puncta in (E). (**G**) Representative fluorescent images of SNB-1::GFP under *unc-25* promoter in the dorsal nerve cord axons from ctrl or *vps-34* RNAi worms at D3 and D11. (**H**) Quantification of SNB-1::GFP labeled puncta in (G). A–H, Over 15 worms per group. $n = 2$ independent experiments. Scale bar, 10 μm. Error bars, SEM. $**P < 0.01$, $***P < 0.001$, $****P < 0.0001$. n.s., not significant. All other points as $P > 0.05$. **B, D, F, H**, Unpaired two-tailed Student's *t* test. Raw data can be found in the Supporting information (S1 Data).
(TIF)

**S5 Fig. Improved motor activity in aged *vps-34* KD worms does not depend on autophagy.** (**A**) Representative fluorescent images showing the GFP::LGG-1 in the NR during aging. Orange arrowheads indicate GFP::LGG-1 labeled puncta. (**B**) Quantification of GFP::LGG-1 labeled puncta numbers in the NR. Fifteen worms per group. $n = 3$ independent experiments. (**C and D**) Representative fluorescent images of GFP::LGG-1 in the NR from Ctrl RNAi and *vps-34* RNAi (C) or WT and *vps-34(qx546)* (D). (**E and F**) Quantification of GFP::LGG-1 puncta numbers in Ctrl RNAi, *vps-34* RNAi, WT and *vps-34(qx546)* nerve ring. Fifteen worms per group. $n = 2$ independent experiments. (**G and H**) Motor behavior assays (Body bends) comparing ctrl and autophagy-related gene RNAi groups. (G) VH624 strain was used. (H) wild-type strain was used. (**I**) Frequency of body bends was quantified in autophagy mutants

compared to N2. (**J**) Frequency of body bends was quantified in N2 or autophagy mutants with *vps-34* KD. The fold changes of *vps-34* RNAi vs. control RNAi in each group were calculated: wild-type (1.5-fold) and other autophagy mutant such as *epg-8* (1.5-fold), *atg-9* (1.5-fold), *atg-3* (1.5-fold), *epg-5* (1.8-fold), *atg-2* (1.3-fold). **G** and **H**, 10 worms per group. **I** and **J**, 15 worms per group. *n* = 2 independent experiments. Error bars, SEM. **\*\*P* < 0.01, \*\*\**P* < 0.001, \*\*\*\**P* < 0.0001. n.s., not significant. All other points as *P* > 0.05. **B, E, F, J,** Unpaired two-tailed Student's *t* test. **G–I,** One-way ANOVA with Dunnett's multiple comparisons test. Scale bars, 10 μm. Raw data can be found in the Supporting information (S1 Data).
(TIF)

**S6 Fig. Neuronal endocytosis is not dramatically affected in *vps-34(qx546)*.** (**A**) Representative fluorescent images of YFP::RAB-5 under *unc-25* promoter in the dorsal nerve cord inhibitory neurons in young (D3) and aged (D11) WT or *qx546* worms. (**B**) Quantification of YFP:: RAB-5 puncta number and intensity in D3 and D11 WT or *qx546* worms. (**C**) Representative fluorescent images of GFP::RAB-7 in the ventral nerve cord in young (D3) and aged (D10) WT or *qx546* worms. (**D**) Quantification of GFP::RAB-7 puncta number and intensity in D3 and D10 WT or *qx546* worms. A–D, Over 15 worms per group. *n* = 2 independent experiments. Error bars, SEM. *\*P* < 0.05, \*\**P* < 0.01, \*\*\**P* < 0.001, \*\*\*\**P* < 0.0001. n.s., not significant. **B, D,** Unpaired two-tailed Student's *t* test. Raw data can be found in the Supporting information (S1 Data).
(TIF)

**S1 Table. Positive clone hits from 3 repetitive screen.**
(XLSX)

**S2 Table. GO annotation of 34 positive hits and their partners in S1C Fig.**
(XLSX)

**S3 Table. Lifespan statistics for S2G Fig.**
(XLSX)

**S4 Table. List of *C. elegans* strains.**
(XLSX)

**S5 Table. Expression constructs.**
(XLSX)

**S6 Table. Primers sequences.**
(XLSX)

**S1 Data. Numerical data and statistical analysis for Figs 1C, 1D, 1E, 1F, 1G, 1H, 1I, 1J, 1K, 1L, 2C, 2D, 2E, 2F, 2G, 2I, 3B, 3C, 3D, 3E, 3F, 3G, 3H, 3I, 3K, 3L, 3M, 4B, 4C, 4D, 4E, 4F, 4J, 4K, 4M, 4O and S1A, S1B, S1D, S1E, S1F, S1G, S1H, S1I, S1J, S2C, S2D, S2E, S2F, S2G, S2H, S2I, S2K, S2L,S2N, S2O, S2P, S2Q, S3A, S3B, S3C,S3D, S4B, S4D, S4F, S4H, S5B, S5E, S5F, S5G, S5H, S5I, S5J, S6B, and S6D.**
(XLSX)

**S1 Raw Images. Images of complete blots.**
(PDF)

## Acknowledgments

We thank Dr. Xiaochen Wang and Dr. Hong Zhang at Institute of Biophysics (Chinese Academy of Sciences) for providing *vps-34(qx546)* and autophagy mutants, Xiajing Tong at

University of Science and Technology for providing stain TXJ1245, Drs. Hong Zhang, Xiaochen Wang for critically reading the manuscript, and Xin Li and Bin Chen for technical support. Several *C. elegans* strains used in this work were provided by the Caenorhabditis Genetics Center (CGC).

## Author Contributions

**Conceptualization:** Yan Zhang.

**Data curation:** Zhongliang Hu, Yamei Luo, Yuting Liu, Yaru Luo, Liangce Wang, Yuling Peng, Rui Wei, Shangbang Gao, Yan Zhang.

**Formal analysis:** Zhongliang Hu, Yamei Luo, Yuting Liu, Yaru Luo, Liangce Wang, Yuling Peng, Rui Wei, Shangbang Gao.

**Funding acquisition:** Shangbang Gao, Yan Zhang.

**Investigation:** Zhongliang Hu, Yamei Luo, Yuting Liu, Shangbang Gao, Yan Zhang.

**Methodology:** Zhongliang Hu, Yamei Luo, Yuting Liu, Shangbang Gao.

**Project administration:** Shangbang Gao, Yan Zhang.

**Resources:** Shengsong Gou, Da Jia, Yuan Wang, Shangbang Gao, Yan Zhang.

**Supervision:** Shangbang Gao, Yan Zhang.

**Validation:** Shangbang Gao, Yan Zhang.

**Writing – original draft:** Yan Zhang.

**Writing – review & editing:** Da Jia, Yuan Wang, Shangbang Gao, Yan Zhang.

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
