## [Editor Report · Decision Letter 0]

13 Jan 2023

Dear Dr Zhang, 

Thank you for submitting your manuscript entitled "Partial inhibition of Class III PI3K VPS-34 ameliorates motor aging and prolongs health span" for consideration as a Research Article by PLOS Biology.

Your manuscript has now been evaluated by the PLOS Biology editorial staff as well as by an academic editor with relevant expertise and I am writing to let you know that we would like to send your submission out for external peer review.

However, we would like to consider the manuscript as a Short Report, which have only 4 main figures. Thus, please reduce the number by making some of the main figures supplementary. When you add the metadata (see below), please select Short Report as a type of article from the drop down menu.

Before we can send your manuscript to reviewers, we need you to complete your submission by providing the metadata that is required for full assessment. To this end, please login to Editorial Manager where you will find the paper in the 'Submissions Needing Revisions' folder on your homepage. Please click 'Revise Submission' from the Action Links and complete all additional questions in the submission questionnaire.

Once your full submission is complete, your paper will undergo a series of checks in preparation for peer review. After your manuscript has passed the checks it will be sent out for review. To provide the metadata for your submission, please Login to Editorial Manager (https://www.editorialmanager.com/pbiology) within two working days, i.e. by Jan 17 2023 11:59PM.

Kind regards,

Ines

--

Ines Alvarez-Garcia, PhD

Senior Editor

PLOS Biology

---

## [Decision Letter · Decision Letter 1]

15 Mar 2023

Dear Dr Zhang,

Thank you for your patience while your manuscript entitled "Partial inhibition of Class III PI3K VPS-34 ameliorates motor aging and prolongs health span" went through peer-review at PLOS Biology as a Short Report. Please also accept my apologies for the delay in providing you with our decision. Your manuscript has now been evaluated by the PLOS Biology editors, an Academic Editor with relevant expertise, and by two independent reviewers.

The reviews are attached below. As you will see, the reviewers find the conclusions of your manuscript interesting and worth pursuing for publication. Nevertheless, both ask for several clarifications, missing controls and further details about the methods used. Reviewer 1 also thinks you should repeat some of the experiments using a higher number of individuals to confirm the results.

In light of the reviews and after consulting with the Academic Editor, we are pleased to offer you the opportunity to address the comments from the reviewers in a revision that we anticipate should not take you very long. We will then assess your revised manuscript and your response to the reviewers' comments with our Academic Editor aiming to avoid further rounds of peer-review, although might need to consult with the reviewers, depending on the nature of the revisions.

We expect to receive your revised manuscript within 1 month, but please email us (plosbiology@plos.org) if you have any questions or concerns, or you think you would need more time for the revision.

**IMPORTANT - SUBMITTING YOUR REVISION**

3. Resubmission Checklist

a) *Ethics Statement*

We also require the license number in your Ethics Statement.

b) *PLOS Data Policy*

Please note that as a condition of publication PLOS' data policy (http://journals.plos.org/plosbiology/s/data-availability) requires that you make available all data used to draw the conclusions arrived at in your manuscript. If you have not already done so, you must include any data used in your manuscript either in appropriate repositories, within the body of the manuscript, or as supporting information (N.B. this includes any numerical values that were used to generate graphs, histograms etc.). Please also INDICATE IN EACH FIGURE LEGEND WHERE THE DATA CAN BE FOUND. For an example see here: http://www.plosbiology.org/article/info%3Adoi%2F10.1371%2Fjournal.pbio.1001908#s5

We require the data underlying the graphs shown in the following figures:

Fig. 1C-L; Fig. 2C-G, I; Fig. 3B-I, K, M; Fig. 4B-F, J, K, M, O; Fig. S1A, B, D-H; Fig. S2C-I; Fig. S3A-D; Fig. S4B, D, F, H; Fig. S5B, E, F-J and Fig. S6B, D

d) *Blurb*

Please also provide a blurb which (if accepted) will be included in our weekly and monthly Electronic Table of Contents, sent out to readers of PLOS Biology, and may be used to promote your article in social media. The blurb should be about 30-40 words long and is subject to editorial changes. It should, without exaggeration, entice people to read your manuscript. It should not be redundant with the title and should not contain acronyms or abbreviations. For examples, view our author guidelines: https://journals.plos.org/plosbiology/s/revising-your-manuscript#loc-blurb

e) *Published Peer Review*

Sincerely,

Ines

--

Ines Alvarez-Garcia, PhD

Senior Editor

PLOS Biology

Reviewers' comments

Rev. 1:

In this work, Hu et al. performed a novel RNAi screen to identify genes whose inactivation allows worms to maintain mobility in old age. They identified and further characterized the role of one of the candidate genes (out of a total of 34) that encodes the class III Phosphatidylinositol 3-kinase, VPS-34. Genetic and electrophysiological approaches led them to conclude that VPS-34 reduces neurotransmission in aged neurons via inhibition of the conversion of PI3P to PI4P. The use of a drug targeting this kinase in mice leads to an increase in the size of muscle fibres and a change in their metabolism at an advanced age.

These results are interesting because the molecular actors responsible for motorneurons aging in C.elegans are largely unknown. The authors used multiple approaches relevant to the characterization of their phenotype, but some results are still preliminary and need further substantiation:

- the number of worms counted for the motility tests is insufficient (double counting and blind-scoring are required)

- in the current manuscript, PI3P or PI4P scoring is based on poor quality and unconvincing images of the nerve ring (which vesicle size was considered? in which ROI (region of interest); why not always make the measurements in the nerve cord as in Figure 3 L/M (in which the number of dots should be given relative to the length of the cord)

-In dual RNAi experiments (Figure 3D) the control should be vps-34 RNAi mixed 1:1 with HT115 control bacteria rather than concentrated vps-34 RNAi

-Normalisation of RT-QPCR data should use 3 control genes instead of one.

-What is the impact of the vps-34 mutation on longevity ?: no values and statistics are reported for figure S2G

-Figure S5: the presentation of the results should compare the effect of vps-34 RNAi in a wild-type or mutant background (rather than control RNAi or vps-34 in a mutant background)

- the legend in figure 3H is incomplete (strain? age?)

-a thrashing test for all candidates to confirm their phenotype would be welcome

Rev. 2:

In this manuscript, the authors established a simple and handy paradigm to assess motor ability in aging C. elegans. By performing genome-wide RNAi screening, the authors identified 34 genes that contribute to motor aging. One of the top hits, Class III PI3K VPS-34, modulates motor aging by regulating neurotransmission in motor neurons and the integrity of muscles. Genetic and pharmacological inhibition of VPS-34 with SAR405 ameliorate age-related motor decline in both worms and mice. In this study, Zhang and her colleges provide a unique platform to explore motor aging, and find a novel mechanism underlying motor aging. The conclusions in this study are compelling, and the findings are of high significance in field of motor aging. There are some minor issues:

1. It's interesting that partial inhibition rather than complete blockade of VPS34 could improve motor ability. Are there any deleterious effects with VPS-34 inhibition?

2. VPS34 is the only class III PI3K regulating the PI(3)P level and it may be required in multiple organs or cell types, so whether the inhibition also has protective effect on other organs?

3. What's the physiological function of VPS-34, and whether it's function is age-dependent?

4. The data showed no significant difference in D5 and D7 RNAi on delaying motor decline, is there a key time point to inhibit the VPS-34?

---

## [Editor Report · Decision Letter 2]

5 May 2023

Dear Dr Zhang,

Thank you for your patience while we considered your revised manuscript entitled "Partial inhibition of Class III PI3K VPS-34 ameliorates motor aging and prolongs health span" for publication as a Short Report at PLOS Biology. This revised version of your manuscript has been evaluated by the PLOS Biology editors and the Academic Editor.

Based on our Academic Editor's assessment of your revision, we are likely to accept this manuscript for publication, provided you satisfactorily address the data and other policy-related requests stated below.

In addition, not sharing the data is against our journal policy and, in response to Rev. 2 Point 8 request of performing thrashing test for all candidates to confirm their phenotype, you mention that "Since our studies on some of these genes are either under review elsewhere or currently ongoing, we regret we can’t provide a detailed list of these results due to conflict of interest concerns." Thus, please include the data, otherwise you will have to remove the statement entirely.

We expect to receive your revised manuscript within two weeks. 

*Published Peer Review History*

*Press*

Sincerely,

Ines

--

Ines Alvarez-Garcia, PhD

Senior Editor

PLOS Biology

ETHICS STATEMENT:

Thank you for providing the ethics statement. Please also include the approval/license number.

Fig. 1C-L; Fig. 2C-G, I; Fig. 3B-I, K, M; Fig. 4B-F, J, K, M, O; Fig. S1A, B, D-H; Fig. S2C-I; Fig. S3A-D; Fig. S4B, D, F, H; Fig. S5B, E, F-J and Fig. S6B, D

BLURB

Please also provide a blurb which (if accepted) will be included in our weekly and monthly Electronic Table of Contents, sent out to readers of PLOS Biology, and may be used to promote your article in social media. The blurb should be about 30-40 words long and is subject to editorial changes. It should, without exaggeration, entice people to read your manuscript. It should not be redundant with the title and should not contain acronyms or abbreviations. For examples, view our author guidelines: https://journals.plos.org/plosbiology/s/revising-your-manuscript#loc-blurb

We require the original, uncropped and minimally adjusted images supporting all blot and gel results reported in an article's figures or Supporting Information files. We will require these files before a manuscript can be accepted so please prepare and upload them now. Please carefully read our guidelines for how to prepare and upload this data: https://journals.plos.org/plosbiology/s/figures#loc-blot-and-gel-reporting-requirements

---

## [Editor Report · Decision Letter 3]

13 May 2023

Dear Dr Zhang,

Thank you for the submission of your revised Short Report entitled "Partial inhibition of Class III PI3K VPS-34 ameliorates motor aging and prolongs health span" for publication in PLOS Biology. On behalf of my colleagues and the Academic Editor, Josh Dubnau, I am delighted to say that we can in principle accept your manuscript for publication, provided you address any remaining formatting and reporting issues. These will be detailed in an email you should receive within 2-3 business days from our colleagues in the journal operations team; no action is required from you until then. Please note that we will not be able to formally accept your manuscript and schedule it for publication until you have completed any requested changes.

PRESS

Sincerely, 

Ines

--

Ines Alvarez-Garcia, PhD

Senior Editor

PLOS Biology
